# The Influence of Fatigue Loading on the Durability of the Conveyor Belt

Nikola Ilanković ⬤, Dragan Živanić * and Ninoslav Zuber ⬤

Faculty of Technical Sciences, University of Novi Sad, Trg Dositeja Obradovića 6, 21000 Novi Sad, Serbia
* Correspondence: zivanic@uns.ac.rs

**Abstract:** The conveyor belt is by its structure a textile composite. As a load-supporting element of the conveyor, the belt withstands variable loads during its operations. In order to investigate the influence of the level and variability of loading on the life of the belt, tests were carried out on specimens in laboratory conditions. A testing device was specially designed and made for these tests that enabled precise control and monitoring of the loading as well as number of loading cycles up to fracture. This research provides an overview of the influence of fatigue loading on the fatigue life of the belt. The methodology of the conducted research is explained with a description of important technical parameters of the testing device. A physical experiment and a corresponding numerical simulation using the FEM method were carried out with multiple loading levels of belt specimens. Based on the obtained results, appropriate conclusions were made; at loads less than 70% of the breaking strength, the lifetime of the belt is very long. Attention was drawn to additional influences that could not be covered by the experiment and possible directions for further research were indicated.

**Keywords:** conveyor belt; textile composite; fatigue assessment; durability

## 1. Introduction

Conveyors are devices that transport material without stopping for loading and unloading. One of the most important elements of any conveyor is its belt. The role of the belt is to transport the material, accept the impact energy at the point of the material loading, withstand the temperature and chemical effects of the transported material, and withstand the nominal loading throughout its working life. The belt can be seen as a textile composite consisting of an upper rubber protective layer, a loading carrying layer (carcass), and a lower rubber protective layer. The role of rubber layers is to protect the carcass from the impact of the transported material [1], while the carcass has a multiple role:

- It needs to provide adequate ultimate tensile strength with respect to loading due to transporting material;
- It must absorb the impact energy of the material being loaded;
- It has to ensure the longitudinal and transverse rigidity of the belt in order to maintain its shape during the transport of the material.

The layers are bonded by a vulcanization process, so that, during assembly, the ends of the belt are joined by a hot or cold vulcanization process or with mechanical connectors.

The carcass is made of one or more layers of woven textile material. The layers are obtained by weaving warp threads that extend in the longitudinal direction and weft threads that extend in the transverse direction. The thread material can be of natural or artificial origin. The most common combination used is polyester-nylon, where polyester is used for the warp threads due to its good mechanical characteristics, while nylon is used for the weft threads because it is more elastic than polyester and behaves better when the belt bends laterally due to leaning on a set of support rollers [2].

Due to its inhomogeneous structure and the nature of the loading that changes during one transport cycle, the conveyor belt is a subject of study that is very relevant today.

Research is carried out to better understand the behavior of the belt during exploitation in order to eliminate unnecessary damage or breakage of the belt due to its incorrect selection.

As for the influencing factors on the life of the belt, the main cause of belt deterioration is the contact between the belt and the transported material, as well as between the belt and the support elements of the conveyor. Damage to the belt also occurs due to a combination of various factors—the influence of sun, rain, snow and ice, chemical reagents, and mechanical effects. Therefore, it is difficult to precisely define the exact cause when belt damage occurs [3]. In terms of mechanical effects, the belt is affected by malfunctioning parts of the conveyor such as support idlers and pulleys. Also, damage can occur during the use of machinery that is required during the transport and installation of the belt [4]. The main cause of wear of the rubber protective layers is abrasion. Micro-cuts are created that become larger over time, and parts of the rubber layer fall off, which creates a crack that leads to accelerated deterioration of the supporting layer [5]. The loading place is one of the critical points in terms of damage [6]. The most common forms of the manifestation of damage to the belt due to the impact of the material are the appearances of longitudinal cracks, damaged edges, and a thinned protective rubber layer [7]. As already mentioned, environmental influences can lead to premature degradation of the conveyor belt. Conditions such as increased air humidity and temperature affect the material of the carcass differently. Unlike nylon, polyester has a higher moisture absorption rate that leads to the degradation of its mechanical characteristics [8]. Hot vulcanization temperatures in the range of 140 °C to 160 °C allow the vulcanization process to be carried out safely in relation to the mechanical characteristics of the carcass, while temperatures above 200 °C lead to complete degradation [9]. The mechanical properties of conveyor belts deteriorate under long-term exposure to thermal shocks [10].

As the joint of the belt is the weakest point of the belt, much research is focused on examining the behavior of the joint. There are several reasons that lead to the loss of the capacity of the joint. They can be represented by joint technology, structural material, and geometrical features [11]. If the belt breaks, transport flows are disrupted, which leads to production losses. Therefore, solutions are being developed for intelligent monitoring of the condition of the belt and an automatic control system that would be able to stop the operation of the conveyor when significant damage is detected [12]. One of the most important parameters that can be monitored during exploitation is the elongation of the belt. Dynamic condition monitoring of the belt using a neural network model could enable a response from an intelligent belt condition monitoring system [13]. The mechanical resistance of the belt end joint is defined by two components—ultimate tensile strength and fatigue limit. During the tension test, the joint tears without delamination of the layers. On the other hand, during dynamic tension testing, delamination occurs first, i.e., adhesion fracture between the supporting layers occurs first. Therefore, it is important to study the fatigue resistance of adhesion between layers [14]. The fatigue behavior of the joint, i.e., the number of loading cycles until the belt layers delaminate, is affected by the modulus of elasticity of the belt, the ultimate tensile strength of the belt, and the adhesive strength of the belt joint [15].

Testing the belt specimens without a joint, i.e., the belt itself, has been a significant aim of recent research. The elongation of the specimens under the reference loading is influenced by the type of material of the carcass, the number of carcass layers, and the tensile strength of the material of the carcass [16]. The material of the carcass and the number of plies affect the ultimate tensile strength of the belt [17]. During exploitation, the belt loses its mechanical properties. Degradation of tensile strength occurs, flexibility of the belt decreases, while resistance to delamination decreases significantly, in certain cases up to 100% [18]. Innovative methods of monitoring the condition of the belt specimen during experimental tests have been developed. One of the most promising is the method of recording the belt specimen with computer tomography. As a result, a 3D model of the specimen is obtained in which the damage caused during the test can be clearly seen [19]. Through computer tomography, it is possible to monitor the change in the distance between

the fibers of the carcass, which indicates damage, and it is possible to spot critical points in time [20].

## 2. Fatigue of the Conveyor Belt as a Textile Composite

Fatigue testing of textile composites is a relatively new concept when compared to fatigue examining of structural materials [21].

The building fibers of the conveyor belt are semi-crystalline materials which show highly elastic properties—they act partly as a viscous fluid, and partly as an elastic material. The elastic and plastic regions of visco-elastic materials and the corresponding yield point of the material and its limit are not precisely defined for such materials. Fibers are isotropic in nature, but reinforcement is anisotropic, which means that there are moderate constraints on molecular alignment. This feature directly affects the type of fracture due to the application of alternating loading to the fiber. Therefore, fiber fatigue fracture can be far more complex than fatigue fracture experienced by metal. It is not possible to consider any single fracture mode as solely responsible for the ultimate fracture of the fiber. The fiber can fail due to tension, bending, twisting, abrasion, environmental conditions, etc. There are over twenty different fracture mechanisms that have been identified that can lead to ultimate fiber fracture [22]. Fibers used for the carcass of conveyor belts have a slender and long structure. They behave rigidly during tensile tests, while they are very flexible during bending tests. The largest number of fiber tests are for tension [23]. The shape of the fracture that can be observed when the fiber fails due to fatigue can be used during diagnostics and provides insight into the mechanisms that led to the fracture. A comparison of fiber fracture due to tension and fatigue is shown in Figure 1.

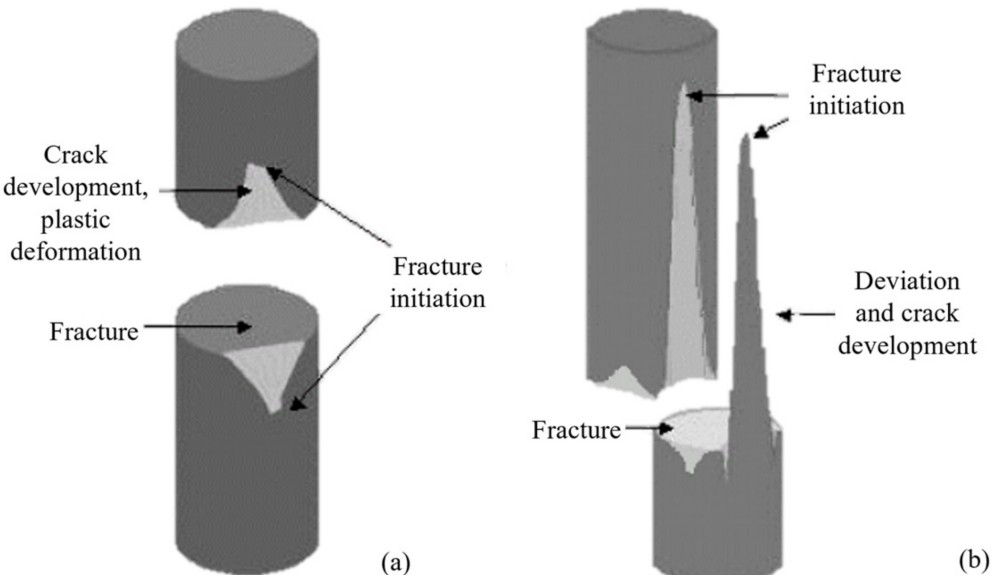

**Figure 1.** Fracture morphology of polyester and polyamide fibers: (**a**) due to tension; (**b**) due to fatigue [24].

When a fiber fractures due to tension, an inclined fracture zone spreads from the place of the initial crack occurrence, which is caused by the plastic deformation of the fiber. During the crack propagation process, the cross-section of the fiber decreases, which leads to the ultimate fracture of the fiber. In the case of fiber fracture due to fatigue, the initial crack appears on the surface of the fiber as in the previous case, but the direction of crack propagation is different. While the crack in tension propagates towards the interior of the fiber, in the fatigue fracture mechanism the crack propagates along the fiber at a small angle to the axis of the fiber. The crack propagates to a critical point when the cross-section of the fiber decreases and damage occurs [24].

In fiber-reinforced composites, which include conveyor belts, the fibers are load-carrying elements, while the matrix has the role of holding the fibers together and protecting them from harmful environmental influences. During testing of textile composites, two categories influence the obtained results: the parameters of the tested material and the parameters of the fatigue test method.

As for the parameters of the tested material, one of the most important is thermal conductivity. When testing specimens at higher frequencies (5–10 Hz), the temperature of the tested object increases to over 100 °C [25]. When the thermal conductivity is low, heat accumulates in the sample, and the material deteriorates faster due to heating. This feature is extremely important when testing specimens of conveyor belts because the upper and lower protective rubber layers act as insulators that additionally lead to heat accumulation. During the manufacturing process of the textile composite, damage may occur that will later affect fracture due to fatigue. Significant in-plane shear may occur, leading to the formation of folds that later negatively affect the fatigue behavior of the specimen [26].

Regarding the parameters of the fatigue test method, it is necessary that the specimen is large enough to include a large number of weaving segments because the inhomogeneity of the material due to the arrangement of the fibers in it significantly affects the fatigue behavior. The specimen should have a protective layer on the ends so that the jaws of the testing device do not damage the base material and lead to early fracturing [27]. Moreover, stress ratio $R$ is very important during testing, and it is defined as follows:

$$R = \frac{\sigma_{MIN}}{\sigma_{MAX}} \tag{1}$$

It is necessary to control the loading amplitude. When the loading amplitude is below a value of tensile strength of the material, the resulting fatigue is called high-cycle fatigue because it occurs after a large number of cycles. When the loading amplitude is high in relation to the tensile strength of the material, the resulting fatigue is called low-cycle fatigue because it occurs after a small number of cycles. The nature of damage that occurs during the test depends on the loading amplitude. Plain tension damage is present in low-cycle fatigue, while typical fatigue damage occurs in high-cycle fatigue [28].

Textile composites have good resistance to fatigue, which is reflected in the length of life in fatigue testing. However, this does not refer to the number of cycles leading to initial damage. The first phase of material weakening due to fatigue is caused by the creation of damage zones. These zones contain a large number of microscopic damage points and the initial delamination between fiber and matrix. It is important to note that the first phase of material weakening occurs very early, already after several hundred cycles. The second phase is characterized by the gradual degradation of the material, which is reflected in the reduction of the stiffness of the material. More significant damage occurs in the third phase—fiber cracking and uncontrolled delamination between fibers and the matrix, which leads to accelerated deterioration and eventual fracture [29]. The degradation process of the textile composite is shown in Figure 2. First, the delamination of the weft threads and the matrix occurs. During this phase, transverse cracking of the weft threads occurs. In the second phase, the delamination spreads along the weft threads and several close delaminations merge into a large delamination. At the same time, there is a small delamination, the so-called meta delamination between the weft and warp threads. In the third phase, i.e., at the end, the cracks propagate from the weft to the warp threads, and large cracks and fatigue damage occur [30].

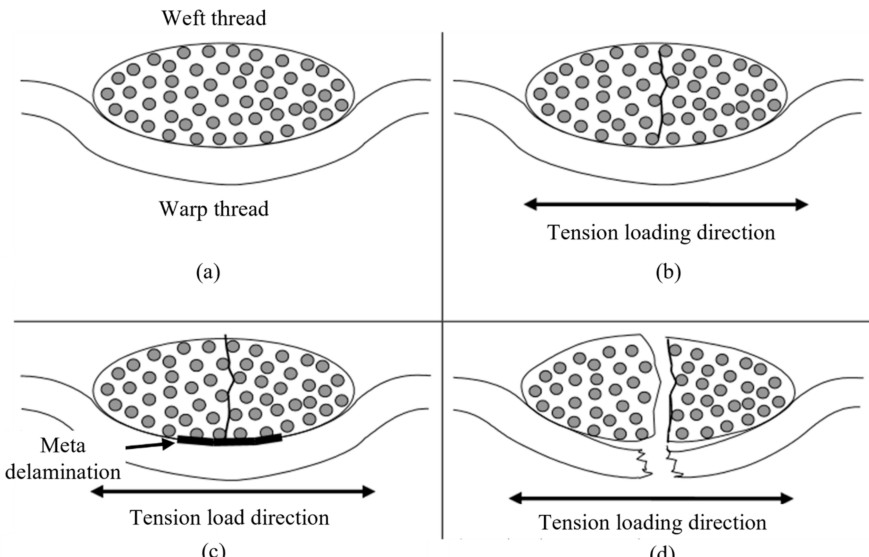

**Figure 2.** Gradual development of textile composite damage due to fatigue: (**a**) initial phase–no failure due to fatigue; (**b**) first damage phase – delamination of weft threads and the matrix, transverse cracking; (**c**) second damage phase – delamination spreads between weft threads and the matrix, meta delamination between weft and warp threads; (**d**) third damage phase – crack propagation from weft to warp threads and fatigue fracture occurs [31].

## 3. Testing and Modeling of Conveyor Belt Fatigue Behavior

The uniaxial tension/tension test is the most commonly used test. Servo-hydraulic test devices or devices with servo-electric motors with adequate drivers are used. Alignment of the specimens in the jaws is extremely important. Bending loading should not occur, so a ball-and-socket joint is used between the jaw and the force sensor in order to transmit only forces and not torques. It is necessary to measure the deformation of the specimen, and for this, high-precision displacement sensors—extensometers—are used, or data from the driver of the servo-electric motor can also be used as displacement sensors. The highest possible test frequency is chosen in order to shorten the duration of the test, but it is necessary to take into account that the increase in the temperature of the specimen due to high frequencies leads to the degradation of the material characteristics. During the test, it is possible to monitor a large number of parameters—loading amplitude, deformation, test frequency, changes in axial stiffness, variations in Poisson's ratio, etc. The change in the axial stiffness can be obtained as the quotient of the measured loading and the measured strain from the displacement sensor. Depending on the type of fiber, the stiffness degradation can range from just a few percent to several tens of percent [32].

By analyzing valid standards concerning testing of specimens of conveyor belts [33], it was determined that there are no direct guidelines regarding fatigue testing of conveyor belt specimens. Therefore, it is possible to apply higher level standards related to tension/tension fatigue testing of composite materials. Those are ASTM D 3479/D 3479M-96 Standard Test Method for Tension-Tension Fatigue of Polymer Matrix Composite Materials [34] and ISO 13003 Fibre-reinforced plastics—determination of fatigue properties under cyclic loading conditions [35] standards. These standards provide general principles of fatigue testing of textile composites and therefore can be applied to the testing of conveyor belt specimens.

There are two technical reasons why modeling fatigue damage in textile composites is difficult and expensive. The first reason is that there are multiple levels at which damage mechanisms are present—micro level (fibers and matrix), medium level (single layer), and macro level (textile composite). The second reason is the impossibility of producing identical specimens due to the nature of the production of textile composites [36].

There are two divisions of existing fatigue models for textile composites.

According to [37], it is possible to divide fatigue damage models according to fatigue criteria:

- Fatigue criterion based on macroscopic strength;
- Fatigue criterion based on residual strength;
- Fatigue criterion based on residual stiffness;
- Fatigue criterion based on actual damage mechanisms.

According to [38], fatigue models for textile composites can be classified into three major categories:

- Fatigue life models that do not consider the actual mechanisms of material deterioration, but are based on S (S—stress)–N (N—number of cycles) or Goodman diagrams;
- Phenomenological models of residual stiffness/strength;
- Progressive damage models that use one or more variables that are associated with measurable damage representations.

Regarding the first category, there are many models based on established S-N diagrams for metals. This approach requires a large number of experiments for each individual material, for different ways of layering, different loading conditions, etc. As for the second category, it consists of phenomenological models of residual stiffness/strength. These models propose evolutionary connections that describe the gradual degradation of the stiffness or strength of the textile composite specimens in terms of measurable characteristics. Residual stiffness models take into account the degradation of the elastic characteristics of materials during fatigue testing. The stiffness of the specimen can be measured frequently or continuously during the fatigue test and can be determined without damaging the specimen. These models can be deterministic where an exact stiffness value is predicted, or they can be statistically evaluated where a distribution of stiffness values is predicted. Residual strength models describe the degradation of the initial static tensile strength during fatigue testing. They were developed from the need to define the remaining life of the material. Statistical models such as the two-parameter Weibull distribution are most often used to describe the residual strength and probability of fracture of a textile composite after a defined number of cycles. As for the third category, it consists of models of progressive damage. These models describe the degradation of the textile composite in direct relation to the specific damage. They relate one or more variables to measurable damage, quantitatively taking into account the progression of the actual damage mechanisms. The most important outcome of all fatigue models is the prediction of fatigue life. Each of the mentioned categories uses its own criteria for determining the moment of ultimate fracture, and thus the fatigue life.

One of the general fatigue life models of textile composites that can be applied to tension/tension testing of conveyor belt specimens is the Bond's variable stress ratio fatigue life model [39]. The S-N diagram is described by the following relation:

$$\sigma_{MAX} = b \log(N) + c \tag{2}$$

where:

- $\sigma_{MAX}$—the maximum value of the stress amplitude;
- $N$—number of cycles to fracture;
- $b$ and $c$—fourth-order polynomials in function of an arbitrary function $R''$. For the tension/tension loading case, the stress ratio is $0 < R < 1$, while the arbitrary function is calculated as $R'' = 4 + R$.

Stated fourth-order polynomials in case of variable stress ratio $R \neq const.$ are calculated as follows:

$$b = K(R'')^4 + L(R'')^3 + M(R'')^2 + N(R'') + P \tag{3}$$

$$c = Q(R'')^4 + T(R'')^3 + U(R'')^2 + V(R'') + W \tag{4}$$

## 4. Materials and Methods

The aim of the research was to analyze the influence of fatigue loading on the lifetime of the belt. In this sense, the belt specimen was subjected to variable loading in order to determine the number of cycles that the belt specimen can withstand before fracture. In doing so, belt specimens were loaded with different values of the mean loading in relation to the breaking strength of the belt and within defined limits of the minimum and maximum loading. This test approach simulates the loading of the belt in operation because the force in the belt, during movement along the conveyor route, is variable within certain limits. Belt specimens were subjected to loading in the form of a sinusoidal function, where each period represented one work cycle. For example, a 2-km-long conveyor belt moving at a speed of 1.68 m/s takes 2381 s for one working cycle, which is represented in the experiment by one period of the specimen loading. In this way, for the given example, the annual operation of the belt, which amounts to 13,245 working cycles, is simulated by an identical number of specimen-loading periods.

### 4.1. Method

Testing was carried out on a testing device specially designed for this purpose, UZITT MKM 5000. The testing device is located in the laboratory at the Faculty of Technical Sciences in Novi Sad, Serbia, Figure 3.

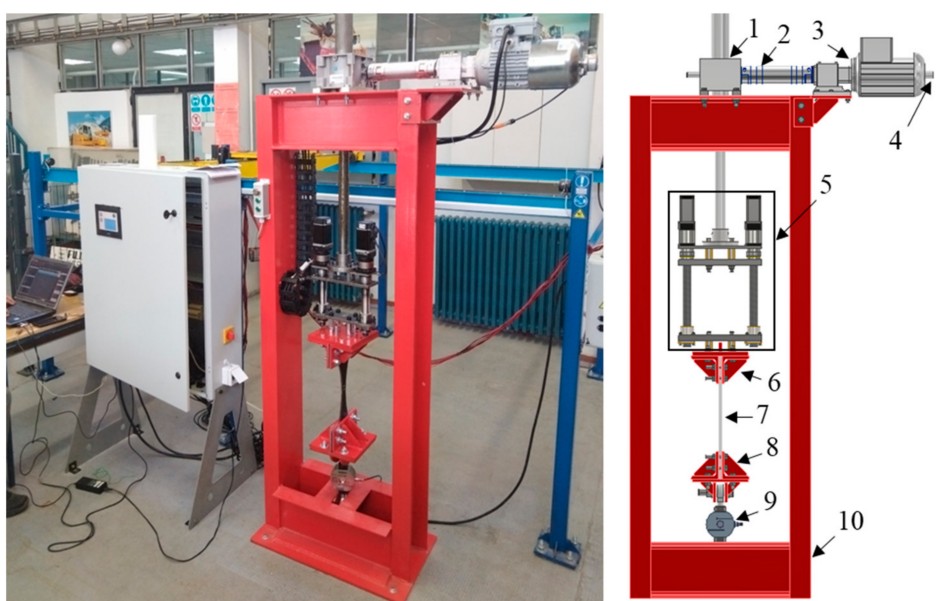

**Figure 3.** Device for testing conveyor belts UZITT MKM 5000 during the experiment: worm gear screw jack (1), connecting shaft (2), helical geared electric motor (3), rotary encoder (4), fatigue testing module (5), upper jaw (6), belt specimen (7), lower jaw (8), force cell (9) and the supporting frame (10).

The characteristics of the testing device are as follows:

- Maximum loading for static testing: 50,000 N;
- Maximum loading for dynamic testing: 25,000 N;
- Maximum travel during static testing: 700 mm;
- Maximum travel during dynamic testing: 200 mm;
- Maximum speed of static tensioning: 100 mm/min;
- Maximum speed of dynamic tensioning: 16 mm/s.

The fatigue testing module is shown on Figure 4. It is demountable, so it can be removed if static tests are carried out over 25,000 N, which is its nominal loading capacity.

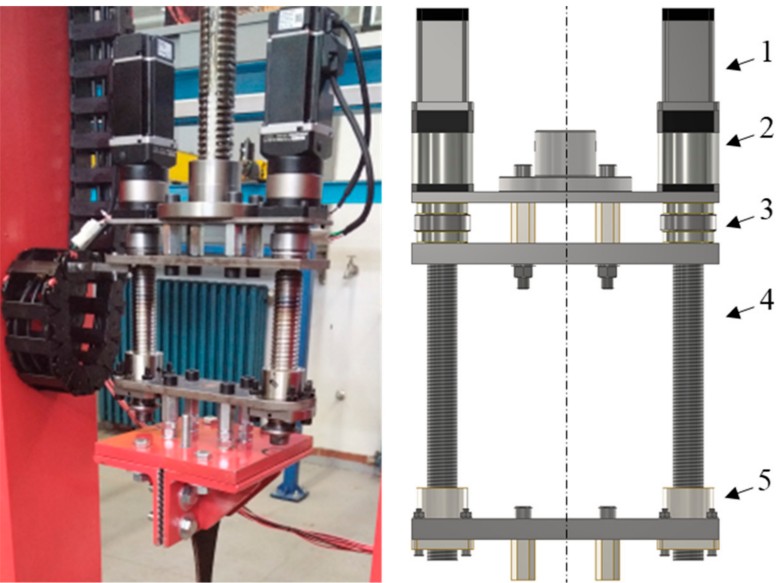

**Figure 4.** Fatigue testing module: servo motor (1), planetary gearbox (2), bearing (3), threaded spindle (4), nut with roller balls for the threaded spindle (5).

As for the static test, the displacement of the upper jaw is measured via a rotary encoder. The tensioning speed is defined by the speed of the AC motor connected to the Hitachi NES1-007 HBE variable frequency controller (Hitachi, Tokyo, Japan). The frequency controller receives the set value of the speed via an analog signal from the Fatek PBS MC 64 PLC (Fatek, New Taipei, Taiwan), which has an FBS 2D 4A analog signal generation module.

The signal from the tension force cell goes to the measuring amplifier module FBS LC in the PLC.

As for the dynamic test, the displacement of the upper jaw and the tensioning speed are defined through the PLC that controls the Fatek SD3 servo drivers that regulate the operation of the servo motors. The user sets the test parameters through the PLC, and the PLC, with the help of the servo driver and internal algorithm, establishes the measurement parameters (amplitude, frequency, signal profile, mean force). The PLC in communication with the servo drivers manages measurements and monitors measured results. It then directs them to a PC that is used as a user interface for setting parameters and recording results. In order for both servo motors to move identically, the PLC controls one servo driver in terms of setting the signal shape, frequency, amplitude, etc., and the second servo driver copies the movement of the first one. For this purpose, software that has a module for static testing and a module for dynamic testing is specially designed.

The connection between the PLC and the PC is based on the RS485 interface. The baud rate is 19,200 bits per second. A similar protocol is used for communication between the PLC and the servo driver, as well as between the PLC and the touch panel. This communication is resistant to interference and ensures a sufficient amount of data in time, for this particular test.

### 4.2. Material

A new conveyor belt was used for testing with a carcass which was created by plain weaving. The warp threads are made of polyester, while the weft threads are made of nylon. This type of textile is known as EP in conveyor belts. The model of the belt carcass is shown in Figure 5. The mark of the belt is EP 500/3 Y 526858.

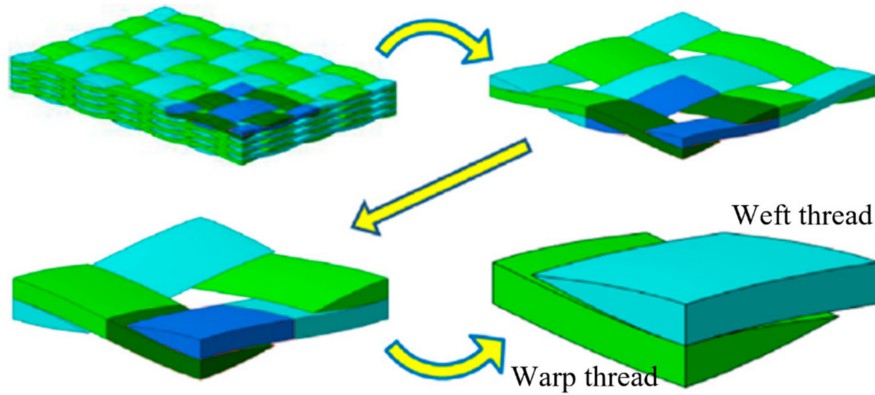

**Figure 5.** Model of a plain-woven EP conveyor belt carcass [40].

The belt characteristics are provided in Table 1.

**Table 1.** Characteristics of the selected belt from the manufacturer's datasheet.

| Characteristic | Unit | Min | Max | Value |
|---|---|---|---|---|
| Breaking strength longitudinally | N/mm | 500 | - | 523 |
| Elongation at break | % | 12 | - | 29.5 |
| Working elongation—10% | % | - | 1.5 | 1.34 |
| Adhesion top cover/1st ply | N/mm | 3.5 | - | 4.4 |
| Adhesion 1st ply/2nd ply | N/mm | 5 | - | 7.6 |
| Adhesion 2nd ply/3rd ply | N/mm | 5 | - | 6.8 |
| Adhesion bottom ply/bottom cover | N/mm | 3.5 | - | 4 |
| Tensile strength of cover | N/mm$^2$ | 20 | - | 20 |
| Elongation of cover | % | 400 | - | 522 |
| Abrasion resistance | mm$^3$ | - | 130 | 122 |

Specimens were manufactured according to EN ISO 285 [41]. Figure 6 shows the specimen scheme and the sampling process. The width of the specimen at the narrowest part is 25 mm. The maximum force that the belt specimen can withstand is 13,075 N.

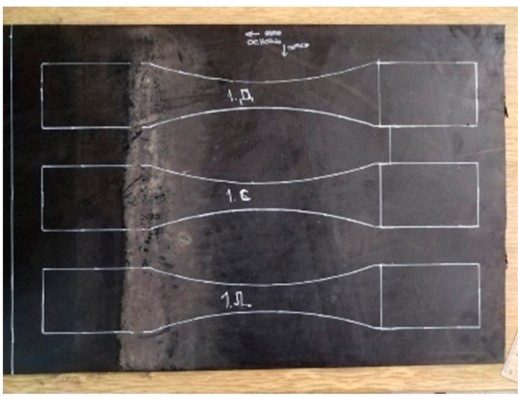

**Figure 6.** Specimens of conveyor belt EP 500/3 Y 526858.

## 5. Results and Discussion

The test was performed with five levels of the mean loading (10 kN, 9.6 kN 8.64 kN, 8.48 kN and 7.68 kN) where its minimum and maximum values ranged within ±25% from the mean loading, which achieved the loading ratio of $R = 0.6$. In this way, the maximum loading reached values of 95.6%, 91.7%, 82.6%, 81%, and 73.4% in relation to the breaking strength of the belt specimen (13,075 kN). Ten repetitions were performed for all five loading levels.

Measuring devices, control automation, and adequate preparation of the experiment enabled the conditions for the experiment to be done correctly:

- There was no slipping of specimens in jaws, and the specimens were placed coaxially with the axis of the device;
- Ball-and-socket joints attached to the tension-loaded cell provided only axial force transfer;
- Ambient temperature and air humidity were monitored so that the experiment would not take place in inadequate conditions;
- The test frequency was 1 Hz in order to avoid additional heat generation in the specimen that could affect results.

Figure 7 shows the testing of the specimen in three phases—the initial state, during the test, and the final phase, i.e., the specimen breaking.

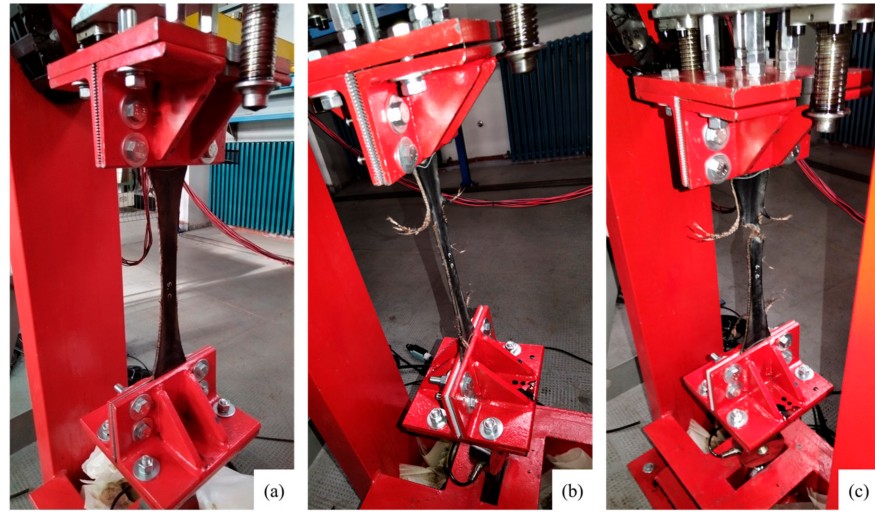

**Figure 7.** Phases of specimen testing: (**a**) initial phase, i.e., undamaged specimen; (**b**) during testing, visible damage; (**c**) the final stage, i.e., specimen breaking.

Table 2 shows the test results.

**Table 2.** Results.

|  | Loading Level | | | | |
|---|---|---|---|---|---|
|  | I | II | III | IV | V |
| Test force mean [kN] | 10 | 9.6 | 8.64 | 8.48 | 7.68 |
| Test force max. [kN] | 12.5 | 12 | 10.8 | 10.6 | 9.6 |
| Percentage of breaking force [%] | 95.6 | 91.7 | 82.6 | 81 | 73.4 |
| Test force min. [kN] | 7.5 | 7.2 | 6.48 | 6.36 | 5.76 |
| Number of specimens | 10 | 10 | 10 | 10 | 10 |
| Average no. of cycles until fracture | 823 | 3168 | 39,180 | 82,472 | 467,568 |
| Standard deviation in number of cycles | 190 | 384 | 2978 | 4619 | 28,521 |
| Relative standard deviation [%] | 22.9 | 12.1 | 7.6 | 5.6 | 6.1 |

As the results were used at a force ratio $R = 0.6$, and since it is the belt tearing force that is important for conveyor belts, and not the tension, a transformation of the fatigue life model given in Equation (2) was made:

$$F_{MAX} = b\ln(N) + c \qquad (5)$$

where:

- $F_{MAX}$ [N]—maximal testing force;
- $N$—number of cycles to fracture.

Coefficients *b* and *c* were calculated from the curve obtained by interpolation based on experimental results. Their value is *b* = −451.2 and *c* = 15,584. The curve equation, i.e., the curve of the F-N diagram for specimens of the new belt EP 500/3 Y 526858, is

$$F_{MAX} = -451.2\ln(N) + 15,584 \tag{6}$$

Figure 8 shows the results and the F-N curve graphically.

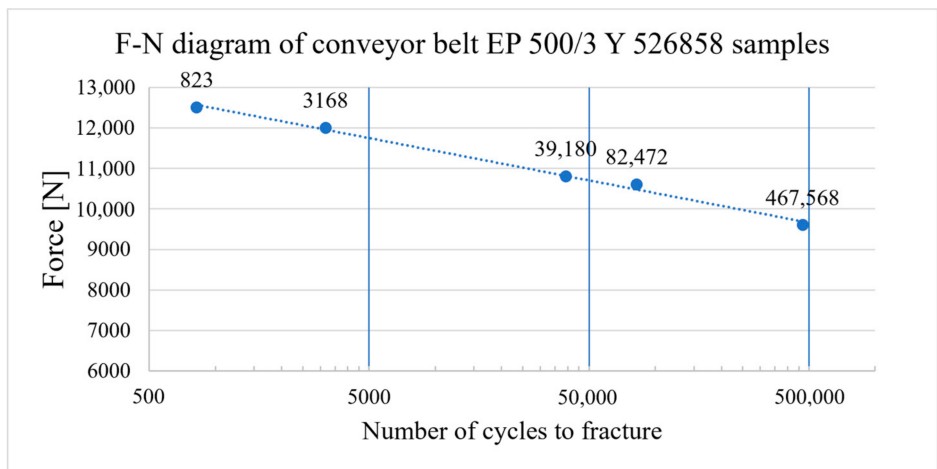

**Figure 8.** Graphically presented results and F-N curve.

Figure 9 shows some of fractured specimens after fatigue tests.

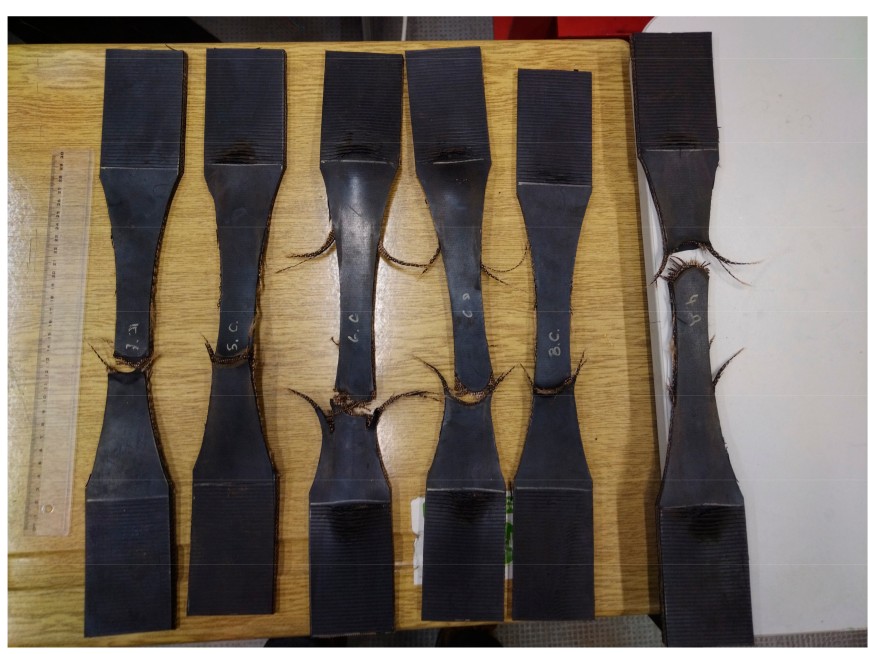

**Figure 9.** Some of the fractured specimens after fatigue tests.

Concerning the fatigue limit, in [42] it is stated that the maximum value of number of cycles for fiber-reinforced composites until fracture that should be considered is 10,000,000. By introducing this number into the Equation (6), the maximum force required for specimen fracture after 10,000,000 cycles would be 8311.5 N, i.e., 63,5% in relation to the breaking strength of the belt specimen:

$$F_{MAX} = -451.2\ln(10,000,000) + 15,584 = 8311.5 \text{ N} \tag{7}$$

In order to verify the obtained experimental results, a FEM model of the belt specimen was created. The results are shown in Figure 10.

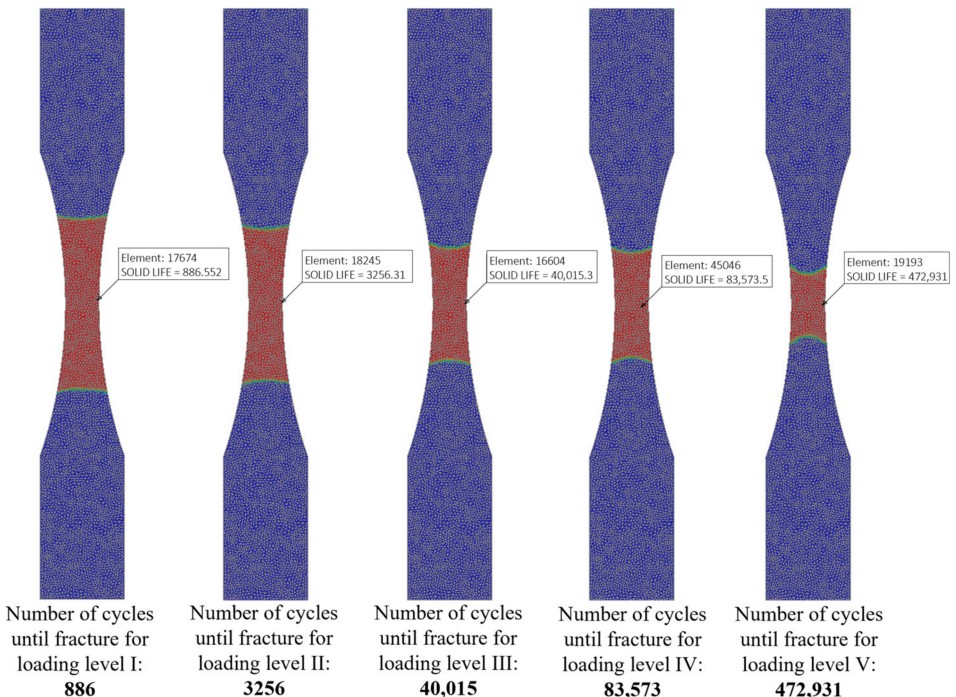

**Figure 10.** Results of FEM fatigue analysis.

As for the FEM model, first the idealization was done. The belt specimen was formed as a solid model and was prepared by removing the rubber protective layers that would only distort the results. The model was defined based on the material of the belt carcass. The multi-axial fatigue module was activated in the Autodesk Inventor Nastran 2022 software to perform the fatigue test. Second, boundary conditions were applied. Fixed constraint was applied at the top surface of the specimen, while the loading was applied at the bottom surface of the specimen. In order to simulate the sinusoidal loading cycle, loading history table data were created. Loading scale factor in the range of $0.6 \div 1$ was entered for 1 s in order to achieve 1 Hz test frequency. After that, mesh was applied. Local mesh control was applied on the edges of the specimen. Finally, the simulation was started. During the first iteration of the simulation of the first specimen, analysis was done with default mesh size. After that, the simulation was done with the half of the default mesh size. It took several iterations to achieve result convergence. The final mesh size was 2 mm. The number of finite elements was 61,244.

Numerical simulations were carried out on the formed belt specimen model in an identical manner to the physical experiment described in the previous part of the paper.

The obtained results of the FEM analysis were compared with the average values of the experimentally obtained results, and both are shown in Table 3.

**Table 3.** Results of comparative analysis of experimental and FEM results.

|  | Loading Level | | | | |
|---|---|---|---|---|---|
|  | I | II | III | IV | V |
| Average no. of cycles until break during physical experiments | 823 | 3168 | 39,180 | 82,472 | 467,568 |
| No. of cycles until fracture according to FEM | 866 | 3256 | 40,015 | 83,573 | 472,931 |
| Percentage of result difference [%] | 5 | 2.7 | 2.1 | 1.2 | 1.1 |

The obtained value of the average number of cycles until the belt breaks at the maximum loading of 95.6% of the nominal breaking strength, only 823 cycles, indicates that the damage occurs at the first loading cycles. This is due to the fiber damage mechanisms explained previously. In metals, molecular bonds enable homogeneity of the structure, and such material can withstand a greater number of cycles at a maximum loading close to the nominal ultimate tensile strength than is the case with textile composites such as conveyor belts.

At the maximum loading level of 91.7% of the nominal belt-breaking strength, the number of cycles did increase (823 → 3168), but that number still does not ensure an economically profitable and acceptable lifetime of the belt.

Moving to the maximum loading levels of 82.6% and 81%, there was a significant increase in the number of cycles to fracture. It should be noted here that the higher number of cycles to fracture (39,180–82,472) with a slight decrease in the loading level (82.6–81%) indicates the sensitivity of the considered influence, which requires further and more detailed analysis.

Also, a very significant increase in the number of cycles to fracture (467,568) is observed at a loading level of 73.4%, which may indicate a potentially very long life of the belt at a lower loading level of 70% of the nominal breaking strength.

For the example that is mentioned earlier, the tested belt would work for 4 months at a maximum loading level of about 90% of the nominal breaking strength, at 80% for about 6 years, and at 75% for about 30 years.

It must be noted that the previous calculation only takes into account damage to the belt specimen due to tension/tension fatigue, i.e., it does not take into account the actual damage to the belt during exploitation, which occurs due to the bending of the belt around the pulleys, the impact of the material on the belt during loading, misalignment with the direction of the conveyor, improper tensioning of the belt, etc.

## 6. Conclusions

The paper presents basic postulates of testing textile composites such as conveyor belts on fatigue damage mechanisms. The mechanisms that lead to fatigue damage are explained and are clearly differentiated from the mechanisms that lead to damage in static testing. Fatigue testing of conveyor belt specimens under tension/tension loading type is presented. Specimens were tested at a constant loading ratio $R$ on a specially designed UZITT MKM 5000 device specially for this occasion.

Based on the obtained results, it was determined that the number of cycles that the belt specimen can reach before fracture significantly depends on the loading.

In order for the obtained results to be practically applied, in the sense of choosing the optimal belt according to the real loading and a certain configuration of the belt conveyor, it would be necessary to verify the way of representing the working cycle of the belt as one period of the form of a sine function. Verification would be possible by testing belt material, after a known number of cycles, and comparing them with the results on specimens of new belts with the same characteristics.

As for further research directions, it is necessary to carry out experiments with variable loading ratio $R$ in order to obtain a more general mathematical model. Also, it is necessary to carry out tests of conveyor belts with different numbers and materials of carcass layers in order to examine their influence on the fatigue life.

To get a more complete picture of the fatigue behavior of conveyor belts, further research is needed to examine the influence of atmospheric conditions such as temperature and humidity and also damage due to the impact of materials on the belt on the fatigue life. It is necessary to test the specimens with different test frequencies (within the limits of the allowed temperature in the specimen) in order to analyze the influence of the tension speed on the life of the specimen. The relationship between the initial dimension of mechanical damage and the lifetime of the belt should be defined, and the influence of the direction of the critical zones in relation to the axis of the belt—longitudinal and transverse—should be

examined. Inhomogeneities and defects in the belt can locally lead to a significant reduction in the capacity of the belt, which can cause a significant shortening of its working life.

The difficulty in conducting the described tests is the total time required for the experiment. At the used test frequency of 1 Hz, it would take 2 years to test 10 specimens up to 6,000,000 cycles, which is, based on the life curve defined in this work, the limiting number of working cycles at 65% of the nominal breaking strength of the belt specimen, which is why the same was not performed. Therefore, it is necessary to determine the highest possible test frequency that does not negatively affect the mechanical characteristics of the specimen in order to shorten the duration of the experiment.

**Author Contributions:** Conceptualization, N.I. and D.Ž.; methodology, N.I. and D.Ž.; software, N.I. and D.Ž.; validation, N.I., D.Ž. and N.Z.; formal analysis, N.I.; investigation, N.I.; resources, N.I.; data curation, N.I.; writing—original draft preparation, N.I. and D.Ž.; writing—review and editing, N.I., D.Ž. and N.Z.; visualization, N.I. and D.Ž.; supervision, N.I., D.Ž. and N.Z.; project administration, N.I.; All authors have read and agreed to the published version of the manuscript.

**Funding:** This research received no external funding.

**Institutional Review Board Statement:** Not applicable.

**Informed Consent Statement:** Not applicable.

**Data Availability Statement:** The data presented in this study are available upon reasonable request from the corresponding author. The data are not publicly available due to privacy issues.

**Conflicts of Interest:** The authors declare no conflict of interest.

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
