# Peer review of "The Influence of Fatigue Loading on the Durability of the Conveyor Belt"

_applsci, doi:10.3390/app13053277_

Round 1
Reviewer 1 Report
Abstract,
The sentence “Based on ob-20 tained results, appropriate conclusions were made” is too general. Include relevant results in the abstract.
Keywords, I’m not sure working life is a correct translation. You can use Duration of use, Service life, product lifetime.
“The most common combination used is polyester-nylon “ Provide citation.
“which is most commonly used for weft threads, polyester, which is 71 most commonly used for warp threads “ this information has been already written in line 46.
In equation 1 R_MIN reads as R_MN
“Stress boundary limits σMIN and σMAX are defined with algebraic signs”. Please revise the sentence, I do not understand the context or the meaning.
Usually I define the loads in N instead of kg.
Table 1. D o the authors know the standard deviations for the values?
Figure 6. The cutting zone can be critical here. Have the authors revised the damage caused during specimen cutting?
In table 2 the standard deviation is referred to the average n of cycles?
More detail on the element used to perform the FEM, refining, size… loads, restrictions and were are applied is needed,
Author Response
Dear reviewer,
We wish to express our deepest gratitude for your revision of our manuscript. Please see the attachment for the response. We apologize in advance if we missed any comment. We hope that we met your expectations.
Sincerely,
Autors

Reviewer 2 Report
Dear Authors,
The paper follows some technical problems with experimental and numerical approaches. It sounds good, nevertheless, some corrections are required.
All comments are presented in the paper after revision.
Reviewer

Author Response
Dear reviewer,
We wish to express our deepest gratitude for your thorough revision of our manuscript. Please see the attachment for the response. We apologize in advance if we missed any comment. We hope that we met your expectations.
Sincerely,
Autors

Round 2
Reviewer 1 Report
The authors have answered all my doubts
Author Response

(The authors gave the same response as above.)

Reviewer 2 Report
Dear Authors,
I have reviewed the paper with corrections from the authors and I do not have any comments or suggestions. The article is ready for publication in the Applied Sciences.
Best regards,
Reviewer

Author Response

(The authors gave the same response as above.)
